# Preparation of a Hydrophobic-Associating Polymer with Ultra-High Salt Resistance Using Synergistic Effect

**DOI:** 10.3390/polym11040626

**Published:** 2019-04-04

**Authors:** Yang Zhang, Jincheng Mao, Jinzhou Zhao, Xiaojiang Yang, Tao Xu, Chong Lin, Jinhua Mao, Hongzhong Tan, Zhaoyang Zhang, Bo Yang, Shaoyun Ma

**Affiliations:** 1State Key Laboratory of Oil and Gas Reservoir Geology and Exploitation, Southwest Petroleum University, Chengdu 610500, China; yangzhang10234@163.com (Y.Z.); 15827742134@163.com (T.X.); linc0616@163.com (C.L.); 18328085244@163.com (J.M.); tanhz163@163.com (H.T.); zhangzy78yx@163.com (Z.Z.); 201611000098@stu.swpu.edu.cn (B.Y.); 2Shanghai King Materials Industry Limited Liability Company, Shanghai 201700, China; msy08007@163.com

**Keywords:** hydrophobic-associating water-soluble polymers, salt resistance, synergistic effect, mechanism

## Abstract

Polymer, SRP-2-1, was synthesized by micellar polymerization and characterized by ^1^H NMR. Salt tolerance and viscoelasticity tests verified that the salt resistance of SRP-2-1 was promoted by the synergistic effects of oxyethylene groups, sulfonate, and hydrophobic chains. It is suggested that the structure of SRP-2-1 became more compact with increasing salinity. Furthermore, a mechanism is proposed as to why SRP-2-1 solution has excellent salt-resistance properties. The experimental results indicate that, because of the good shear resistance properties, the polymer SRP-2-1 could be used as an alternative in many fields, for instance in fracturing fluids, enhanced oil recovery, and sewage treatment.

## 1. Introduction

Partially hydrolyzed polyacrylamide (HPAM) is widely used in various fields, such as enhanced oil recovery, water treatment, and agriculture, as a result of its availability in large quantities and low cost [1]. However, HPAM has poor temperature and salt resistance, meaning that it cannot meet the requirements for reservoir stimulation under high-temperature and high-salt conditions [2,3,4]. In other words, the collapse of the polyacrylamide tends to occur at higher temperatures and salinities, which greatly limits the application of PAM in oil fields, especially in EOR or fracking [5]. Hence, there is an urgent need for researchers to develop a water-soluble polymer with good temperature and salt resistance properties.

Currently, polymers with temperature and salt resistance are effectively prepared by introducing a structural unit with special functions into the copolymers [6,7]. Specifically, there are five methods to improve the properties of polymers from the perspective of molecular structure: (i) Improving the thermal stability of the polymer backbone by introducing a rigid group which increases the rigidity of the molecular chain [8,9]. (ii) Improving the salt resistance of the polymer by introducing large side groups or rigid groups, such as hydrophobic long chains and sulfonic acid [10]. (iii) Improving the salt resistance of the polymer by introducing functional structural units with the functions of inhibiting hydrolysis, complexing high-valence metal ions, and with strong hydration capacity, such as polyoxyethylene [11]. (iv) Preparation of a polymer with a specific molecular structure in solution due to hydrogen bonds between the macromolecular groups, Van der Waals forces, and hydrophobic associations, etc. (v) Improving the salt resistance and enhancing the viscosity-increasing ability of the polymer by the existence of a crosslinked structure, which could increase the rigidity and difficulty of conformational transformation of the polymer [12].

Based on the information detailed above, a pentamer polymer was designed by copolymerizing acrylamide (AM), acrylic acid (AA), 2-Acrylamido-2-methylpropane sulfonic acid (AMPS), and two kinds of nonionic monomer, *N*,*N*-di-*n*-dodecylacrylamide (DiC_12_AM) and 3,6,9,12,15,18,21,24,27,30,33,36,39,42,45,48,51,54,57,60,63,66,69-tricosaoxahenoctacontyl methacrylate (MAA-EO_23_C_12_). Why were these two functional monomers introduced into the polymer? We considered that there is a twin-tailed long hydrophobic chain on the DiC_12_AM, which could make the molecular chain more rigid and structurally regular [13]. Additionally, there is a hydrophobic backbone (–CH_2_–CH_2_–) of EO groups on the MAA-EO_23_C_12_, which can also significantly increase the viscosity of the polymer solution through repulsion between the hydrophobic and hydrophilic groups [14]. In addition, sulfonic acid was introduced into the polymer by copolymerizing the AMPS with other monomers. There were three reasons for using AMPS: (i) The equilibrium constant of the sulfonic acid group is much larger than that of the carboxylic acid group, keeping the viscosity of the polymer high in brine [15,16]. (ii) The rigidity of the polymer chain was increased by the strong ionization of the sulfonic acid group, ensuring that the polymer chain was stretched in brine [17]. (iii) The sulfonic acid is less shielded by the inorganic salt since it is weakly alkaline [18], meaning that the polymer solution has a higher viscosity retention rate at higher salinity. In addition, micellar polymerization was employed to prepare the polymer in this study so that the hydrophobic chain was distributed in the form of micro-blocks on the polymer macromolecular backbone, which could considerably reduce the entropy resistance when hydrophobic microdomains were formed between hydrophobic groups [19]. This would result in a significant decrease in the critical association concentration of the polymer and would strengthen the polymer thickening ability.

The polymer has excellent thickening ability and thermal stability due to the synergistic effects of the hydrophobic associations, complexation reactions, hydrogen bonds, and Van der Waals forces in high-salinity reservoirs. It is of significant interest to investigate polymers with good salt resistance, which could be used as alternatives for the development of oil and gas resources in deep wells.

## 2. Experimental Section

### 2.1. Materials

Acrylamide (AM), acrylic acid (AA), 2-acrylamide-2-methylpropanesulfonic acid (AMPS), sodium dodecyl sulfate (SDS), 2,2′-azobis (2-methylpropionamide) dihydrochloride (V50), sodium hydroxide (NaOH), sodium chloride(NaCl), potassium chloride(KCl), calcium chloride hexahydrate (CaCl_2_) and magnesium chloride(MgCl_2_) were all purchased from Chengdu Kelong Chemical Reagents Corporation (Chendu, China). HPAM (M_w_ = 16~18 × 10^6^) were purchased from the the Hengju Oil Field Chemical Reagents Co., Ltd. (Beijing, China). Deionized (DI) water was obtained from a water purification system. All chemicals and reagents were utilized without further purification.

### 2.2. Synthesis

#### 2.2.1. Preparation of Monomer

DiC_12_AM and MAA-EO_23_C_12_ were synthesized as follows according to a previously published method [20,21] (Scheme 1 and Scheme 2).

#### 2.2.2. Synthesis of Copolymer

Appropriate amounts of AM (10.0 g), AA (3.0 g), AMPS (2.0 g), MAA-EO_23_C_12_ (0.6 g), and DiC_12_AM (0.1 g) were placed in deionized water, the total monomer concentration was maintained at 30 wt %. Then, the pH was adjusted to 7.0 using NaOH and SDS (1.5 g) was slowly added to the solution. After the SDS was completely dissolved, V50 solution was added using a syringe; the amount of V50 was 0.035 wt % of the total mass of the monomers. Meanwhile, the solution was placed in a UV light fixture (T5 8W UVB) at 25 °C. Polymer colloid was prepared via illumination reaction after 8 h. The copolymers were named as SRP-w-p, where w and p denote the molar ratios of MAA-EO_23_C_12_ and DiC_12_AM, respectively. The SRP-w-p synthesis is shown in Scheme 3. The polymer was then cut into small pieces. Subsequently, it was purified by precipitation with ethanol three times. Finally, it was dried in a vacuum desiccator and stored until further use.

### 2.3. ^1^H NMR Characterization

The nuclear magnetic resonance proton spectra (^1^H NMR) of copolymers in D_2_O were measured using a Bruker AVANCE III HD 400 (Bruker, Karlsruh, Germany). The concentration of the polymer solution was 100 mg/L. 

### 2.4. Water Solubility

The water solubility of the copolymers was obtained by a conductivity method. The solution conductivity was measured using a DDS-307^+^ conductivity meter (Chengdu Century Ark Technology Co., Ltd, Chendu, China). Copolymer particle samples were dispersed and dissolved in deionized water at 25 °C. The water solubility curves were obtained using dissolution time, which is defined as the time from the initial adding of the polymer to the solution conductivity stabilizing.

### 2.5. Thickening Performance

Solution viscosities after the addition of 0.01~0.35 wt % copolymer were measured using a HAAKE MARS III (006-1322) rheometer (Haake, Karlsruhe, Germany) under a shear rate of 170 s^−1^ and at a temperature of 25 °C. Then, the relationship curves between viscosity and concentration were acquired.

### 2.6. Salt Tolerance

Specific amounts of copolymer and sodium chloride were dissolved in deionized water and then placed in a water bath at 25 °C for 24 h to ensure complete dissolution. Similarly, salt solutions of the polymer, in which the salt was potassium chloride, calcium chloride, or magnesium chloride, were prepared in the same way. In addition, the copolymer was dissolved in different concentrations of standard brine (2.0 wt % KCl + 5.5 wt % NaCl + 0.45 wt % MgCl_2_ + 0.55 wt % CaCl_2_). The salinity of the standard brine is regarded as 8 × 10^4^ mg/L. Viscometric measurements were performed using the HAAKE MARS III (006-1322) rheometer at a shear rate of 170 s^−1^ and at 25 °C.

### 2.7. Viscoelasticity

Specific amounts of copolymer and inorganic salt were dissolved in deionized water and then placed in a water bath at 25 °C for 24 h to ensure complete dissolution. The viscoelasticity of copolymer in aqueous or brine solutions was measured using an Anton Paar rheometer(MCR302) with CP50-1-SN30644 plate fixture (diameter = 0.099 mm). To ensure consistency of experimental conditions, all samples were measured at 25 °C.

### 2.8. Dynamic Light Scattering (DLS) Measurement

The hydrodynamic radius (*R_g_*) of a polymer can indicate its particle size and the size of polymer aggregates in aqueous solutions. The particle sizes of copolymers were determined by dynamic light scattering with a wide-angle laser light scatterometer (Brookhaven, BI-200SM, Suffolk, NY, USA). Copolymers were dissolved in deionized water to prepare the polymer solutions. The polymer salt solutions were prepared by adding inorganic salt to 500-mg/L copolymer solutions. The temperature for the tests was 25 °C, the laser module was a 532-Na light source, the detection angle was 90°, and CONTIN software was used for the final data analysis.

### 2.9. Morphological Observation

The aggregating morphology of copolymer in aqueous or brine solutions was investigated using an environmental scanning electron microscope (ESEM; Quanta 450, Hillsboro, OR, USA). All of the samples were dried at room temperature and then frozen at −50 °C using liquid nitrogen. The frozen surfaces of the samples were observed with ESEM operating at an accelerating voltage of 20 kV.

### 2.10. Shear Tolerance

The shear resistance of copolymers in aqueous or brine solutions was measured with a shear rate ranging from 7.34 s^−1^ to 1000 s^−1^ at 25 °C for 30 min using the HAAKE MARS III (006-1322) rheometer.

## 3. Results and Discussion

### 3.1. ^1^H NMR of Copolymers

The SRP-w-p structure was confirmed by ^1^H NMR. Figure 1a shows the ^1^H NMR (400 MHz, D_2_O) spectrum of SRP-2-1. The proton signals at 4.70 ppm were assigned to the solvent protons (D_2_O). The proton signals at 0.77–0.80 ppm were from the –CH_3_ in polymer side chains. The proton signals at 1.21 ppm and at 3.56–3.58 ppm were associated with the –CH_2_-CH_2_ protons in DiC_12_AM. The proton signals at 1.21 ppm could be assigned to the –CH_3_ protons in the polymer main chain. The proton signals at 1.41–1.44 ppm were due to the –CH_3_ in AMPS. The proton signals at 1.57–1.62 ppm and at 2.15 ppm were attributed to the CH_2_-CH– in the polymer main chain. The proton signals at 3.63 ppm were due to –CH_2_ in AMPS. The proton signals at 3.96–3.99 ppm were associated with the alkyl in oxyethylene groups. The proton signals at 5.56–6.26 ppm were due to the vinyl at the residual monomers. Similarly, for SRP-0-1 and SRP-2-0 individual peaks were marked with their corresponding structures, as shown in Figure 1b,c. All results verified that the synthesized polymers were consistent with the targets.

### 3.2. Intrinsic Viscosity and Molecular Weight of Copolymers

These three polymers were all prepared into a solution at the concentration (300, 250, 250 mg/L). Then the time of polymer solution passed the Undiluted Ubbelohde viscometer were measured three times in succession at 30 °C, and recorded the time as t, which was obtained taking the average of the three sets of values. It is worth emphasizing that the error should be controlled under 0.2 s. The time (*t*_0_) was obtained as the same manner of polymer using the solution with the concentration of 1 mol/L NaCl. And the relative viscosity (*η_r_*) of different polymer was calculated through Equation (1):(1)$$\eta_{r}=\frac{t}{t_{0}}$$

The relative viscosity (*η_r_*) of different polymer was measured by Undiluted Ubbelohde viscometer at 30 °C. Then, the intrinsic viscosity [*η*] of different polymer was calculated through Equation (2):(2)$$[\eta ]=\frac{(\eta_{r}-1)\sqrt{\frac{2-\ln3}{\eta_{r}-\ln\eta_{r}-1}}-2\sqrt{\frac{\eta_{r}-\ln\eta_{r}-1}{2-\ln3}}}{c\left(\sqrt{\frac{2-\ln3}{\eta_{r}-\ln\eta_{r}-1}}-1\right)}$$

Finally, the viscosity-average molecular weight of different polymer was obtained through the Mark-Houwink-Sakurada Equation (3) [22]. The experimental results were shown in Table 1.
(3)$$[\eta ]=3.73\times 10^{-4}\times {M_{w}}^{0.66}$$

It is obvious that the molecular weight of the SRP-2-0 is the largest, and that of SRP-0-1 and SRP-2-1 is in second and third position, respectively.

### 3.3. Water Solubility

Generally, the solution conductivity increased continuously as the copolymer particle samples were dissolved in DI water. The conductivity of the solution was constant when the copolymer was dissolved completely. Obviously, the higher the rate of conductivity increase, the higher the dissolution rate of the polymer, which indicates that the water solubility of the polymer is better.

As shown in Figure 2, when the copolymers were first added to the water, the conductivity of the solution was small. With time, the polymer continued to dissolve in the water, causing the conductivity of the solution to increase. Finally, the conductivity of the solution tended to be constant, indicating that the copolymer was dissolved completely. The dissolution times of SRP-2-1, SRP-2-0, and SRP-0-1 were 4.99 min, 6.98 min, and 5.98 min, respectively, indicating that these copolymers have excellent water solubility. Moreover, it is clear that the conductivity trends of these three polymers are consistent and the conductivity of these three polymers in water is almost the same. Analysis believes that there are same addition of AM/AA/AMPS in these three polymers. And compared the addition of AM/AA/AMPS, the addition of DiC_12_AM and MAA-EO_23_C_12_ is relatively small. Hence, the conductivity value of the polymer is mainly attributed to the addition of AM/AA/AMPS. However, the introduction of DiC_12_AM and MAA-EO_23_C_12_ greatly promotes the solubility of polymers in water. Meanwhile, the conductivity of the HPAM solution was not stable until 20 min. Therefore, the water solubility of SRP-2-1, SRP-2-0, and SRP-0-1 are far greater than that of HPAM. There are two explanations for these experimental results. (i) The molecular weights of SRP-2-1, SRP-2-0, and SRP-0-1 are lower than that of HPAM. (ii) Hydrophilic groups, such as sulfonate, ethoxyl, and carboxylate, were introduced into SRP-2-1, which made the solubility of SRP-2-1 better than that of HPAM. The fast-dissolving property of the copolymer is conducive to its application.

### 3.4. Thickening Performance

When a water-soluble polymer is hydrated, the viscosity of the polymer solution increases with the increase of concentration. In this section, we investigated the curves of polymer concentration versus the viscosity of polymer aqueous solution. As shown in Figure 3, the four polymers all exhibit the excellent thickening performance of a directly proportional linear relationship in dilute conditions. Moreover, when the polymer concentration was greater than 1000 mg/L, the viscosities of the polymer aqueous solutions increased sharply with the increase of concentration. In addition, it is noteworthy that the viscosities of polymer aqueous solutions containing SRP-2-1 and SRP-0-1 were lower than for SRP-2-0 and HPAM. This is because the molecular weights of SRP-2-1 and SRP-0-1 were lower than those of HPAM and SRP-2-0 when SDS was used as an emulsifier in the polymerization. However, the fluid containing SRP-2-1 had the highest viscosity growth rate when the polymer concentration increased from 1000 mg/L to 3500 mg/L (278%), the fluid containing SRP-2-0 has the lowest viscosity growth rate (171%), and the fluids containing HPAM (202%) and SRP-0-1 (199%) had viscosity growth rates in second and third position, respectively. The experimental data are shown in Table 2. We considered that there were relatively stronger intermolecular associations, which would further strengthen the binding forces, and hydrogen bonding between oxyethylene groups and acrylamino groups, which would enhance the binding forces between the polymer chains in the SRP-2-1 aqueous solution due to the synergistic effects of DiC_12_AM and MAA-EO_23_C_12_, than for SRP-2-0 and SRP-0-1.

### 3.5. Salt Tolerance

Inorganic salts usually play a negative role in the apparent viscosities of polymer solutions, especially calcium and magnesium ions. Figure 4 shows the effects of salt type and concentration on the apparent viscosities of SRP-2-1, SRP-2-0, SRP-0-1, and HPAM solutions. As shown in Figure 4a, the solution viscosities of copolymers can be divided into three phases due to the special structure of copolymers containing the hydrophobic long chain and oxyethylene groups, which is more complicated than the monotonous downward trend of the viscosity of conventional polymers (HPAM). It is clear that the change in viscosity for SRP-2-1 solution is more dramatic than for SRP-2-0 and SRP-0-1 solutions. When the NaCl concentration was less than 25,000 mg/L, the viscosity of the SRP-2-1 solution showed little decrease with the increase of NaCl concentration. Then, the viscosity increased dramatically until the NaCl concentration reached 12.5 × 10^4^ mg/L, increasing from 78 mPa s to 204 mPa s. Further, when the concentration of NaCl reached 25 × 10^4^ mg/L, the viscosity decreased from 204 mPa s to 102 mPa s.

It is noteworthy that the viscosity of SRP-2-1 solution is always larger than the initial viscosity when the concentration of NaCl is greater than 7.5 × 10^4^ mg/L. In addition, it is always larger than the viscosities of SRP-2-0, SRP-0-1, and HPAM solutions. This is because the sodium ions shielded the anions on the polymer side chains, which reduced the electrostatic repulsion between the anions, leading to the polymer main chain becoming curly. From a macro-perspective, the solution viscosity reduced in the initial phase [23]. Meanwhile, the hydrophobic association effect between the molecular chains was enhanced because of solution polarity enhancement. In addition, oxyethylene groups with strong hydration capacity can enhance the inhibiting hydrolysis, accompanied by an increase in the viscosity of the solution. However, the viscosity of solution decreased at high-salt concentrations when the electrostatic shielding effect of salt on molecular chains exceeded the hydrophobic association between molecules. Meanwhile, the carboxylate ions on the HPAM molecule were shielded by the metal cation in the brine, which made the polymer chain curly. Macroscopically, the viscosity of the HPAM solution drops dramatically [24].

Similarly, the effects of CaCl_2_ and MgCl_2_ on the viscosities of polymer solutions were also investigated, and the results are shown in Figure 4b,c. It is clear that the viscosity of the SRP-2-1 solution was greater than for SRP-2-0 and SRP-0-1 with increased of concentrations of calcium chloride and magnesium chloride, indicating that there is a synergistic effect between DiC_12_AM and MAA-EO_23_C_12_. When the concentrations of calcium chloride and magnesium chloride reached 2 × 10^4^ mg/L, the viscosities of SRP-2-1 solutions decreased from 106.5 mPa s to 54 mPa s and from 135 mPa s to 81 mPa s, respectively. In other words, the viscosity retention rates of SRP-2-1 solutions are 50.71% and 60.0%. However, the viscosity of HPAM solution decreased almost to zero when the concentrations of calcium chloride and magnesium chloride reached 7,000 mg/L, suggesting that the polymer SRP-2-1 has excellent salt resistance compared with HPAM.

It is well known that polymers are often dissolved in brine composed of several inorganic salts from an application point of view. Hence, after evaluating the resistance of polymers to a single inorganic salt, it was necessary to evaluate the effect of compound salts (standard brine) on the viscosities of polymer solutions. The experimental results are shown in Figure 4d. Overall, the viscosity curves of copolymers in standard brine show a trend of first decreasing and then rising. When the salinity of standard brine reached 8 × 10^4^ mg/L, the viscosity retention rates of the SRP-2-1 solution reached 85.11%, indicating that SRP-2-1 has excellent salt resistance and application prospects. Similarly, the mechanisms of the effect of salt on the apparent viscosity of SRP-2-1 solution come down to two points: electrostatic shielding and electric double layer compression of the polymer hydration shell [25]. In summary, the polymer SRP-2-1 has excellent salt resistance compared with SRP-2-0, SRP-0-1, and HPAM, because of the synergy between DiC_12_AM and MAA-EO_23_C_12_.

### 3.6. Viscoelasticity

The viscoelasticity of the polymers SRP-2-1, SRP-2-0, and SRP-0-1 were measured using an Anton Paar rheometer (MR302). The variations in storage modulus (*G′*), loss modulus (*G″*), and crossover point (*G_c_*) [26], which were dependent on frequency, are shown in Figure 5.

The viscoelasticity measurements for SRP-2-1 in DI water and brine are shown in Figure 5a. It is clear that the storage modulus is larger than the loss modulus in the linear viscoelastic area, suggesting that polymer SRP-2-1 in 100% standard brine (8 × 10^4^ mg/L) is the classic plastic fluid. However, in the case of the remaining polymer solutions, the storage modulus and the loss modulus increase gradually with the rise in frequency. Meanwhile, the storage modulus is less than the loss modulus in the low-frequency region, indicating that the viscous modulus plays a dominant role. With the further increase of scanning frequency, the storage modulus is greater than the loss modulus when the scanning frequency is greater than the critical frequency, at which point curves *G′* and *G″* cross each other (*G_c_*), which suggests that the storage modulus plays a dominant role. The relaxation time *t_c_* corresponding to the crossing point *G_c_* can be used to describe the solution viscoelasticity. A longer *t_c_* verifies that the solution structure makes a greater contribution to elastic efficiency. According to the calculation results, the relaxation times of different solutions are shown in Table 3. Based on the experimental results, it can be concluded that the effect of salinity on the viscoelasticity of the SRP-2-1 solution is also first reduced and then increased, which is consistent with its effect on viscosity.

The variations in the storage modulus (*G′*) and the loss modulus (*G″*) versus frequency for SRP-2-0 and SRP-0-1 solutions are shown in Figure 5b,c. It is clear that the negative effect of salinity on viscoelasticity for SRP-2-0 and SRP-0-1 solutions is stronger than for SRP-2-1 solution, suggesting that synergism between DiC_12_AM and MAA-EO_23_C_12_ helps to improve the salt resistance of SRP-2-1 solution.

### 3.7. Molecular Dimensions of SRP-2-1

Molecular dimension changes of SRP-2-1 (500 mg/L) molecules in DI water and 12.5% and 25% standard brine (1 × 10^4^ mg/L and 2 × 10^4^ mg/L) were measured by dynamic light scattering, and the results are shown in Figure 6. For the polymer SRP-2-1 dissolved in 12.5% and 25% standard brine, the mean dimensions were 3459.1 nm and 4966.5 nm, respectively, which are both greater than when dissolved in DI water (461.6 nm). Hence, it is concluded that there is an increase of polymer particle size with increasing standard brine concentration.

In addition, it is worth emphasizing that the size distribution of SRP-2-1 in DI water was relatively centralized. While there are two or more peaks in the size distribution curve of SRP-2-1 in standard brine, the one below 1000 nm was due to the intertwining of polymer molecules, and the one above 1000 nm was attributed to the hydrophobic association of polymer molecules [27]. There were three reasons to explain why the double distribution of size becomes larger with the increase of salinity: (i) When the polymer mixed with the brine, the solution polarity was enhanced, which strengthened the hydrophobic association effect between the molecular chains due to the introduction of nonionic hydrophobic monomer, DiC_12_AM. (ii) The oxyethylene groups on the MAA-EO_23_C_12_ can complex with metal ions. (iii) The inorganic salt would adsorb on the surface of the backbone of polymers. Therefore, the molecular dimensions measured at this time were mainly the size of the supermolecule [28], suggesting that interactions between polymer molecules were promoted by inorganic salts.

### 3.8. Microstructure Analysis

The microstructures of polymer SRP-2-1 solutions with different concentrations of inorganic salts were investigated by SEM, and the results are shown in Figure 7. Figure 7a shows that SRP-2-1 in DI water formed a multimolecular aggregation with a spatial network structure resulting from intermolecular forces, hydrophobic associations, hydrogen bonds, and Van der Waals forces. Meanwhile, it is obvious that sodium chloride crystallized at the surface of the backbone of SRP-2-1, as shown in Figure 7b. In addition, it is clear that the main shapes present in the spatial network structure of the polymer were irregular pentagon and oval, both in inorganic salt solution and in DI water, indicating that polymer SRP-2-1 has good salt resistance properties, especially for calcium and magnesium ions, as shown in Figure 7c,d.

Moreover, we also studied the microstructure of polymer SRP-2-1 in standard brine. It was found that inorganic salts also crystallized at the surface of the backbone of SRP-2-1, covering the surface of the polymer structure and making the structure dense and robust, as shown in Figure 7e,f. More importantly, with increasing salinity, the structure of polymer SRP-2-1 became more compact, which was consistent with the results for salt thickening performance. It is suggested that acrylic acid groups can ionize to form negatively charged carboxylate ions after the addition of inorganic salt to the solution, making the polymer chains extend farther into the aqueous solution. Meanwhile, there is a multilayered three-dimensional network structure with different density pores in the polymer solution because of the entanglement of intermolecular chains [29]. This three-dimensional network structure, with major backbone and minor branches, not only supports the polymer chain but also encapsulates a large number of water molecules, leading to deformation resistance of the polymer SRP-2-1. Macroscopically, polymer SRP-2-1 has excellent salt resistance and thickening properties.

### 3.9. Shear Tolerance

The viscosity curves for polymer solutions as a function of shear rate are shown in Figure 8. It is suggested that the viscosities of SRP-2-1 in different mineralized water gradually decreased with increasing shear rate, indicating that these polymer solutions are all pseudoplastic fluids. It is suggested that these curves could be divided into two stages. The curves decreased dramatically at the lower shear rate range of 7.31–200 s^−1^. However, they were relatively stable at shear rates above 200 s^−1^. It is accepted that the intermolecular forces were destroyed, including hydrogen bonds, Van Edward forces, and hydrophobic associations, and even long polymer chains were broken down. Furthermore, the arrangement of molecular chains was aligned in a specific direction. Moreover, in the case of DI water, the viscosity retention rates for SRP-2-1 and HPAM were 30.41% and 11.55%, respectively, as the shear rate ranged from 7.31 s^−1^ to 170 s^−1^. This indicates that the SRP-2-1 solution has excellent shear-tolerance. Meanwhile, when the SRP-2-1 solution contained different concentrations of metal ions, the viscosity retention rates for SRP-2-1 solutions were 39.89%, 29.26%, 25.56%, 31.02%, and 31.59% as the shear rate ranged from 7.31 s^−1^ to 170 s^−1^. It is noteworthy that these retention rates for SRP-2-1 solutions with different concentrations of metal ions were all larger than for HPAM dissolved in DI water, indicating that SRP-2-1 has excellent shear resistance and salt resistance compared with HPAM.

Based on the experimental results, it is clear that the effect of shear rate on the viscosity of SRP-2-1 in different mineralized water was consistence with the results of viscoelasticity and salt resistance. We consider this attributable to the presence of long PEO chain groups and double-tailed hydrophobic groups: the strong hydrophilic force interactions among the chains that surrounded the long PEO chain groups made the molecular chain stretch; the alkyl chain enhanced the hydrophobic association effect between polymer molecules.

### 3.10. Salt Tolerance Mechanism Analysis

A mechanism is proposed as to why SRP-2-1 solution has excellent shear resistance and salt resistance, as shown in Figure 9, which outlines the synergistic effect of DiC_12_AM and MAA-EO_23_C_12_ on the salt tolerance of polymer SRP-2-1. It is known that complexation reactions occur between multivalent metal ions and EO groups by the lone pair of electrons from the oxygen of the oxyethylene group filling the unoccupied orbital from the metal ion, which in turn enhances the hydrophobicity of polar groups with polyether [11,30].

In summary, in saline water, multimolecular aggregation should promote the generation of a polymer solution via complexation, and accordingly promote Ca^2+^/Mg^2+^ resistance. Moreover, the mutual repulsion of the hydrophobic and the hydrophilic groups reduces curling and entanglement in polymer molecular chains. The role of a twin-tailed long hydrophobic chain on the polymer backbone is similar to the support of pier pillars to a pier deck, indicating that the polymer has good thermal stability because of the increased rigidity of the polymer chain. Therefore, the polymer SRP-2-1 exhibited superior rheological properties and salt tolerance because of the synergy of DiC_12_AM and MAA-EO_23_C_12_.

## 4. Conclusions

In order to promote the salt resistance of hydrophobic-associating water-soluble polymers, the polymer SRP-2-1 was designed incorporating the synergistic effects of the oxyethylene groups, sulfonate, and hydrophobic long chains. Based on a series of experiments, nine conclusions were drawn:

(1) Polymer with good salt resistance was synthesized by micellar polymerization and characterized by ^1^H-NMR spectroscopy. Conductivity tests showed that SRP-2-1 had better water solubility than SRP-2-1, SRP-2-0, and HPAM.

(2) Salt tolerance, viscoelasticity and continuous variable shear tests verified that the salt tolerance and shear resistance of SRP-2-1 was promoted by the synergistic effects of oxyethylene groups, sulfonate, and hydrophobic chains. 

(3) It is suggested that the structure of SRP-2-1 became more compact with increasing salinity, which also verified in molecular dimensions and microstructure analysis of SRP-2-1.

(4) A mechanism is proposed as to why SRP-2-1 solution has excellent salt-resistance properties. The experimental results indicate that, because of the good shear resistance properties, the polymer SRP-2-1 could be used as an alternative in many fields, for instance in fracturing fluids, enhanced oil recovery, and sewage treatment.

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
