# Peer review of "Preparation of a Hydrophobic-Associating Polymer with Ultra-High Salt Resistance Using Synergistic Effect"

_polymers, 2019, doi:10.3390/polym11040626_

Reviewer 1 Report

The manuscript entitled "preparation of a novel ultra-high salt resistance hydrophobic-associating polymer using synergistic effect" by Y. Zhang and al. describes the synthesis of a new water-soluble polymer which has excellent salt resistance. Many analysis regarding the salt resistance, and mechanical properties were carried out. The various comonomers of the random copolymer all play a key role in the properties of the polymer designed as an alternative to currently available polymers for fracturing fluids. This strategy has already been reported, as such the term "novel" should not be used in the manuscript.

I would recommend this mansucript to be published after some minor corrections.

Regarding the synthesis of the copolymer, did the authors encounter any issue in dissolving the monomers in water? Did the addition of SDS improve the solubility of the monomers? Can the authors discuss why SDS needs to be "carefully" added? What are the volume of water used and the quantity of monomers used (a wt% or molar concentration would be useful). Was the solution stirred? The UV wavelength ranges from 10 to 400 nm, can the authors be more specific about the wavelength used? Which solvent was used to dissovle the polymer before precipitating it in EtOH? This section lacks many crucial details to enable someone else to repeat the experiment.

Regarding the 1H NMR analysis, can the authors comment on the presence of residual monomers? Would more purification by precipitation be required? Do the authors expect to observe signals from the SDS protons?

Can the authors provide the ratio of the different monomers in the final polymers and the MW of the polymers (for example on line 175, it is stated the MW of the polymers are lower than that of HPAM but there are no values, again on line 187)? A table that summarises these results would be welcomed. A comparison between the theoretical and experimental ratios would be a good addition too.

Can the authors comment on the conductivity curves of the three SRP polymers? Is it expected that the three polymers display the same conductivity?

Regarding the thickening performance, can the authors comment on the synergistic effects? What would these interactions be?

Regarding the salt tolerance, the authors state that three phases can be seen in Figure 4a. Can these three phases be added to the figure (for example by dotted vertical lines). The authors do not provide explanations why the "special structure" of the polymer plays a key role in having these three phases.

Section 3.6: Can the authors comment on the double distribution of size observed for the three experiments? Were these hydrodynamic diameters measured in terms of number, volume, or intensity? Having these three measurements would help understand the presence of these double distributions.

A few typos can be found in the text. A few examples are listed below, but there are more in the text:

- line 43: "N,N" should be in italics, same for "n"

- line 43: the full name of MMA-EO23C12 should be added

- line 79: synthesis (singular form), same for line 81

- line 81: "MAA" with two "A"

- line 89: add after "DiC12AM" the follwoing: ", respectively". This way, it is obvious that "w" stands for MAA-EO23C12 and "p" for DiC12AM

- line 94: this section is about IR and 1H NMR. The title needs to reflect this.

- line 98: 1H NMR is about "proton", not "hydrogen"

- line 103: There is always a space between the temperature and the unit (refer to line 87 for the correct way of writing). Also line 108 and more in the manuscript.

- line 243: "synergy" rather than "synergism"

- figure 7: the text written in blue in the figure cannot be read

- references: typos in ref 3, 6, 8, 10, ...

Author Response

Comment 1: The manuscript entitled "Preparation of a novel ultra-high salt resistance hydrophobic-associating polymer using synergistic effect" by Y. Zhang and al. describes the synthesis of a new water-soluble polymer which has excellent salt resistance. Many analysis regarding the salt resistance, and mechanical properties were carried out. The various comonomers of the random copolymer all play a key role in the properties of the polymer designed as an alternative to currently available polymers for fracturing fluids. This strategy has already been reported, as such the term "novel" should not be used in the manuscript.

Reply: First of all, thank you very much for your good suggestions. As suggested, we have changed the title to "Preparation of a hydrophobic-associating polymer with ultra-high salt resistance using synergistic effect".

Some other problems

We have revised the manuscript point-to-point as the reviewers suggested. It was embarrassed to find many errors and now we have improved them.

Question 1): Regarding the synthesis of the copolymer, did the authors encounter any issue in dissolving the monomers in water? Did the addition of SDS improve the solubility of the monomers? Can the authors discuss why SDS needs to be "carefully" added? What are the volume of water used and the quantity of monomers used (a wt% or molar concentration would be useful). Was the solution stirred? The UV wavelength ranges from 10 to 400 nm, can the authors be more specific about the wavelength used? Which solvent was used to dissolve the polymer before precipitating it in EtOH? This section lacks many crucial details to enable someone else to repeat the experiment.

Answer 1): First of all, thank you very much for your valuable advice. You are right. We should explain it in manuscript.  Herein, we will answer your questions.

Due to the nonionic monomer, N,N-di-n-dodecylacrylamide, is an oil-soluble monomer so that it cannot dissolve in water. Meanwhile, the addition of SDS can improve its solubility because SDS and N,N-di-n-dodecylacrylamide could form a mixed micelle in the water.

It is our inaccurate expression. We should change the "carefully" to "slowly". If the SDS is not slowly added to the solution, a large amount of bubbles will be generated in the beaker, which will play a negative effect the polymerization.

We have already supplied the specific amount of various monomers in the manuscript. We will stop stirring the solution when the monomers dissolved in water completely. We also supplied the type of UV light fixture we used in the polymerization. The purpose of soaking the polymer with ethanol is to remove the unreacted monomers and initiator and there is no solvent used to dissolve the polymer before precipitating it in ethanol.

Appropriate amounts of AM (10.0 g), AA (3.0 g), AMPS (2.0 g), MAA-EO23C12 (0.6 g), and DiC12AM (0.1 g) were placed in deionized water, the total monomer concentration was maintained at 30 wt%. Then, the pH was adjusted to 7.0 using NaOH and SDS (1.5 g) was slowly added to the solution. After the SDS was completely dissolved, V50 solution was added using a syringe; the amount of V50 was 0.035 wt% of the total mass of the monomers. Meanwhile, the solution was placed in a UV light fixture (T5 8W UVB)at 25 °C. Polymer colloid was prepared via illumination reaction after 8 h. The copolymers were named as SRP-w-p, where w and p denote the molar ratios of MAA-EO23C12 and DiC12AM, respectively. The SRP-w-p synthesis is shown in Scheme 3. The polymer was then cut into small pieces. Subsequently, it was purified by precipitation with ethanol three times. Finally, it was dried in a vacuum desiccator and stored until further use.

Question 2): Regarding the 1H NMR analysis, can the authors comment on the presence of residual monomers? Would more purification by precipitation be required? Do the authors expect to observe signals from the SDS protons?

Answer 2): First of all, thank you very much for your good suggestions. In our experiments, the residual monomers had been washed by the ethanol for many times. During this workup, we have determined the 1H NMR analysis for many times. As suggested, we have characterized copolymers which have been multiple purified. The latest 1H NMR spectrum is shown below. In the process, it is embarrassed for us to find some errors about the analysis of 1H NMR of copolymers and we have revised the relevant expressions in the manuscript and marked it in blue. It is clear that there are still some proton signals at 5.56–6.26 ppm which are attributed to the vinyl at the residual monomers. Moreover, the 1H NMR spectrums of acrylamide, acrylic acid, 2-acrylamide-2-methylpropanesulfonic acid, DiC12AM and MAA-EO23C12, were shown below.

In addition, we did not expect to observe the signals from the SDS protons. And the SDS has been removed by the ethanol from latest 1H NMR spectrum.

The SRP-w-p structure was confirmed by 1H NMR. Figure 1a shows the 1H NMR (400 MHz, D2O) spectrum of SRP-2-1. The proton signals at 4.70 ppm were assigned to the solvent protons (D2O). The proton signals at 0.77–0.80 ppm were from the –CH3 in polymer side chains. The proton signals at 1.21 ppm and at3.56-3.58 ppm were associated with the –CH2-CH2 protons in DiC12AM. The proton signals at 1.21 ppm could be assigned to the –CH3 protons in the polymer main chain. The proton signals at 1.41–1.44 ppm were due to the –CH3 in AMPS. The proton signals at 1.57–1.62 ppm and at 2.15 ppm were attributed to the CH2-CH– in the polymer main chain. The proton signals at 3.63 ppmwere due to –CH2 in AMPS. The proton signals at 3.96-3.99 ppm were associated with the alkyl in oxyethylene groups. The proton signals at 5.56–6.26 ppm were due to the vinyl at the residual monomers. Similarly, for SRP-0-1 and SRP-2-0 individual peaks were marked with their corresponding structures, as shown in Figures 1b and 1c. All results verified that the synthesized polymers were consistent with the targets.

Question 3): Can the authors provide the ratio of the different monomers in the final polymers and the MW of the polymers (for example on line 175, it is stated the MW of the polymers are lower than that of HPAM but there are no values, again on line 187)? A table that summarises these results would be welcomed. A comparison between the theoretical and experimental ratios would be a good addition too.

Answer 3): Yes, you are right. Thanks for your useful suggestions. We indeed did not provide the molecular weight of polymers and the ratio of the different monomers. As suggested, we have tested the molecular weight of polymers by the Undiluted Ubbelohde viscometer at 30 °C. In addition, I have refined the synthesis steps of the polymer. This section has also been modified in the manuscript and marked in blue.

3.2. Intrinsic Viscosity and Molecular Weight of Copolymers

These three polymers were all prepared into a solution at certain concentration (300, 250, 250 mg/L). Then the time of polymer solution passed the Undiluted Ubbelohde viscometer were measured three times in succession at 30 °C, and recorded the time as t, which was obtained taking the average of the three sets of values. It is worth emphasizing that the error should be controlled under 0.2 s. The time (t0) was obtained as the same manner of polymer using the solution with the concentration of 1 mol/L NaCl. And the relative viscosity (ηr) of different polymer was calculated through Equation (1): 

The relative viscosity (ηr) of different polymer was measured by Undiluted Ubbelohde viscometer at 30 °C. Then, the intrinsic viscosity [η] of different polymer was calculated through Equation (2):

Finally, the viscosity-average molecular weight of different polymer was obtained through the Mark-Houwink-Sakurada equation (3) [1]. The experimental results were shown in Table 1.

It is obvious that the molecular weight of the SRP-2-0 is the largest, and that of SRP-0-1 and SRP-2-1 is in second and third position, respectively.

Table 1. Intrinsic viscosity and molecular weight of different polymers.

Question 4): Can the authors comment on the conductivity curves of the three SRP polymers? Is it expected that the three polymers display the same conductivity?

Answer 4): First of all, thank you very much for your suggestions. We indeed did not elaborate on the conductivity curves of the three SRP polymers in the manuscript. As suggested, this section has also been modified in the manuscript and marked in blue.

Moreover, it is clear that the conductivity trends of these three polymers are consistent and the conductivity of these three polymers in water is almost the same. Analysis believes that there are same addition of AM/AA/AMPS in these three polymers. And compared the addition of AM/AA/AMPS, the addition of DiC12AM and MAA-EO23C12 is relatively small. Hence, the conductivity value of the polymer is mainly attributed to the addition of AM/AA/AMPS. However, the introduction of DiC12AM and MAA-EO23C12 greatly promotes the solubility of polymers in water.

Question 5): Regarding the thickening performance, can the authors comment on the synergistic effects? What would these interactions be?

Answer 5): First of all, thank you very much for your suggestions. We indeed did not elaborate on the synergistic effects in thickening performance in the manuscript. As suggested, this section has also been modified in the manuscript and marked in blue.

We considered that there were relatively stronger intermolecular associations, which would further strengthen the binding forces, and hydrogen bonding between oxyethylene groups and acrylamino groups, which would enhance the binding forces between the polymer chains in the SRP-2-1 aqueous solution due to the synergistic effects of DiC12AM and MAA-EO23C12, than for SRP-2-0 and SRP-0-1.

Question 6): Regarding the salt tolerance, the authors state that three phases can be seen in Figure 4a. Can these three phases be added to the figure (for example by dotted vertical lines). The authors do not provide explanations why the "special structure" of the polymer plays a key role in having these three phases.

Answer 6): Thank you very much for your suggestion. We have added two dotted vertical lines to divide the solution viscosities of copolymers into three phases in the Figure 4a. As suggested, we have explained the effect of "special structure" on these three phases. And this section has also been modified in the manuscript and marked in blue.

It is clearly that the solution viscosities of copolymers can be divided into three phases due to the special structure of copolymers containing the hydrophobic long chain and oxyethylene groups, which is more complicated than the monotonous downward trend of the viscosity of conventional polymers (HPAM).

Meanwhile, the hydrophobic association effect between the molecular chains was enhanced because of solution polarity enhancement. In addition, oxyethylene groups with strong hydration capacity can enhance the inhibiting hydrolysis, accompanied by an increase in the viscosity of the solution.

Question 7): Section 3.6: Can the authors comment on the double distribution of size observed for the three experiments? Were these hydrodynamic diameters measured in terms of number, volume, or intensity? Having these three measurements would help understand the presence of these double distributions.

Answer 7): First of all, thank you very much for your suggestions. There may be some problems in my description here, so we have reanalyzed the Molecular Dimensions. As suggested, this section has also been modified in the manuscript and marked in blue.

There were three reasons to explain why the double distribution of size becomes larger with the increase of salinity: (i) When the polymer mixed with the brine, the solution polarity was enhanced, which strengthened the hydrophobic association effect between the molecular chains due to the introduction of nonionic hydrophobic monomer, DiC12AM.(ii) The polyoxyethylene group on the MAA-EO23C12 can complex with metal ions. (iii) The inorganic salt would adsorb on the surface of the backbone of polymers. Therefore, all these three effects made the double distribution of size enlarged.

It is embarrassed for us to find this error. These hydrodynamic diameters of polymer were measured in terms of intensity, not weight percentage. And we have modified the icons of ordinate in Figure 6. Yes, you are corrected. We examined the presence of these double distributions by observing the peak intensity of the polymer at different particle sizes.

Question 8): A few typos can be found in the text. A few examples are listed below, but there are more in the text:

1): line 43: "N,N" should be in italics, same for "n"

2): line 43: the full name of MMA-EO23C12 should be added

3): line 79: synthesis (singular form), same for line 81

4): line 81: "MAA" with two "A"

5): line 89: add after "DiC12AM" the follwoing: ", respectively". This way, it is obvious that "w" stands for MAA-EO23C12 and "p" for DiC12AM

6): line 94: this section is about IR and 1H NMR. The title needs to reflect this.

7): line 98: 1H NMR is about "proton", not "hydrogen"

8): line 103: There is always a space between the temperature and the unit (refer to line 87 for the correct way of writing). Also line 108 and more in the manuscript.

9): line 243: "synergy" rather than "synergism"

10): figure 7: the text written in blue in the figure cannot be read

11): references: typos in ref 3, 6, 8, 10, ...

Answer 8): First of all, thank you very much for your suggestions. It is embarrassed for us to find this error. We have modified these typos and marked in blue in the manuscript.

After careful revision, it is embarrassed that we have revised several minor errors from the manuscript, details as follows:

1) line 77: “monomer” should be “Monomer”.

2) line 97: “H NMR” should be “1H NMR”.

3) line 116: “(2.0% KCl+5.5% NaCl+0.45% MgCl2+0.55% CaCl2)” should be “(2.0 wt% KCl+5.5 wt% NaCl+0.45 wt% MgCl2+0.55 wt% CaCl2)”.

4) line 125: “DLS Measurement” should be “Dynamic Light Scattering (DLS) Measurement”.

5) line 143: “H NMR of Copolymers” should be “1H NMR of Copolymers”.

6) line 197: “investigate” should be “investigated”.

7) Some other expressions have also been modified and marked in blue color in the manuscript. 

Reviewer 2 Report

This paper attempts to  promote and design  the salt resistance of hydrophobic-associating water-soluble polymers. Very interesting paper and I recommend this paper to be published in this journal but after implementing following minor comments 

Introduction needs to be expanded to provide a bit more details in application of polymer in EOR and other applications and also their limitation , example you may include this paper : 

  Studying the Effectiveness of Polyacrylamide (PAM) Application in Hydrocarbon Reservoirs at Different Operational Conditions

Energies2018,11(9), 2201

All figures need to be updated with error bars if possible 

Conclusions should be 3 to 4 bullet points , current style is not appropriate 

Otherwise good work , well done 

Author Response

The response to reviewer 2

After careful revision, we have revised some mistakes and errors from the manuscript. Furthermore, we will thank Prof. Dr. Yang Hu from Texas Tech University, US for helpful discussions and English modification. The revisions are marked in blue color in the manuscript. The following is the justifications:

Comment 1: This paper attempts to promote and design the salt resistance of hydrophobic-associating water-soluble polymers. Very interesting paper and I recommend this paper to be published in this journal but after implementing following minor comments.

Reply: First of all, thank you very much for your comments on this manuscript. We have revised the manuscript as your suggestion. A point by point response to your comments have been listed below.

Question 1): Introduction needs to be expanded to provide a bit more details in application of polymer in EOR and other applications and also their limitation, example you may include this paper: Studying the Effectiveness of Polyacrylamide (PAM) Application in Hydrocarbon Reservoirs at Different Operational Conditions, Energies 2018,11(9), 2201.

Answer 1): Dear reviewer, your suggestion is reasonable that it is significant for the promotion of my paper. That paper is important for the application of polyacrylamide polymers in oil fields. As a suggestion, we have expanded the introduction by quoting that paper whichproviding the application of polymer in EOR and other applications and also their limitation. This section has been marked in blue in the manuscript.

In other words, the collapse of the polyacrylamide tends to occur at higher temperatures and salinities, which greatly limits the application of PAM in oil fields, especially in EOR or fracking.

Question 2): All figures need to be updated with error bars if possible.

Answer 2): Yes, your suggestion is reasonable. As a suggestion, I will use the error bars to update the figures in my future manuscripts and I believe it is significant for the promotion of my paper.

Question 3): Conclusions should be 3 to 4 bullet points, current style is not appropriate.

Answer 3): Yes, your suggestion is reasonable that it is significant for the promotion of my paper. As suggested, we have modified the conclusions to 4 bullet points in the manuscript according to your advice, which are marked in blue.

(1) Polymer with good salt resistance was synthesized by micellar polymerization and characterized by 1H-NMR spectroscopy. Conductivity tests showed that SRP-2-1 had better water solubility than SRP-2-1, SRP-2-0, and HPAM.

(2) Salt tolerance, viscoelasticity and continuous variable shear tests verified that the salt tolerance and shear resistance of SRP-2-1 was promoted by the synergistic effects of oxyethylene groups, sulfonate, and hydrophobic chains.

(3) It is suggested that the structure of SRP-2-1 became more compact with increasing salinity, which also verified in molecular dimensions and microstructure analysis of SRP-2-1.

(4) A mechanism is proposed as to why SRP-2-1 solution has excellent salt-resistance properties. The experimental results indicate that, because of the good shear resistance properties, the polymer SRP-2-1 could be used as an alternative in many fields, for instance in fracturing fluids, enhanced oil recovery, and sewage treatment.

After careful revision, it is embarrassed that we have revised several minor errors from the manuscript, details as follows:

1) line 77: “monomer” should be “Monomer”.

2) line 97: “H NMR” should be “1H NMR”.

3) line 116: “(2.0% KCl+5.5% NaCl+0.45% MgCl2+0.55% CaCl2)” should be “(2.0 wt% KCl+5.5 wt% NaCl+0.45 wt% MgCl2+0.55 wt% CaCl2)”.

4) line 125: “DLS Measurement” should be “Dynamic Light Scattering (DLS) Measurement”.

5) line 143: “H NMR of Copolymers” should be “1H NMR of Copolymers”.

6) line 197: “investigate” should be “investigated”.

7) Some other expressions have also been modified and marked in blue color in the manuscript.
